# Work Efficiency and Economic Efficiency of Actual Driving Test of Proton Exchange Membrane Fuel Cell Forklift

**DOI:** 10.3390/molecules27154918

**Published:** 2022-08-02

**Authors:** Zi’ang Xiong, Haikun Zhou, Xuewen Wu, Siew Hwa Chan, Zhiyong Xie, Dai Dang

**Affiliations:** 1Powder Metallurgy Research Institute, Central South University, Changsha 410083, China; reynoldhsiung@163.com (Z.X.); wucole@163.com (X.W.); 2School of Chemical Engineering, University of New South Wales, Sydney, NSW 2052, Australia; hector.zhou@unsw.edu.au; 3School of Mechanical and Aerospace Engineering, Nanyang Technological University, Nanyang 639798, Singapore; mshchan@ntu.edu.sg; 4School of Chemical Engineering and Light Industry, Guangdong University of Technology, Guangzhou 510006, China

**Keywords:** PEMFC forklift, work efficiency, economic efficiency, PEMFC power system, voltage fluctuation, control strategy, membrane electrode assemblies, power density

## Abstract

A 3.5 tonne forklift containing proton exchange membrane fuel cells (PEMFCs) and lithium-ion batteries was manufactured and tested in a real factory. The work efficiency and economic applicability of the PEMFC forklift were compared with that of a lithium-ion battery-powered forklift. The results showed that the back-pressure of air was closely related to the power density of the stack, whose stability could be improved by a reasonable control strategy and membrane electrode assemblies (MEAs) with high consistency. The PEMFC powered forklift displayed 40.6% higher work efficiency than the lithium-ion battery-powered forklift. Its lower use-cost compared to internal engine-powered forklifts, is beneficial to the commercialization of this product.

## 1. Introduction

The global goal of decarbonization has prompted the exploration of hydrogen energy. Electricity can be generated through renewable energy, however, renewable energy sources are unstable and intermittent during generation which leads to electricity being difficult to apply continuously and stably [1,2]. To tackle this issue, and to greatly improve the utilization rate and stability of renewable energy, employment of hydrogen energy should be considered. The application of hydrogen energy in industrial vehicles is a very promising replacement of conventional power sources [3,4]. Forklifts have been widely applied in manufacturing and warehousing industries for the handling of materials. Existing forklifts are powered by diesel internal combustion engines, lead-acid batteries, lithium-ion batteries and fuel cells. Diesel internal engines have advantages such as high efficiency, short refueling time, strong power, etc., but this is at the expense of exhaust pollution, high working noise, vibration, and labor intensity, which are not eco-friendly or user-friendly, thus requiring research into and the development of better products. One solution to these customer demands is the development of an electric forklift that combines power sources and electric motors.

However, lithium-ion batteries and lead-acid batteries also have problems such as short run time, long charging time, and unstable working efficiencies [5], etc. By contrast, proton exchange membrane fuel cells (PEMFCs) could combine the advantages of both a charging battery and internal combustion engine, whose stable operation at full power, short refueling time, green by-product (water), quiet working environment, and low-temperature operational capability make it an ideal power source for high-performance electric forklifts [6,7,8,9]. PEMFCs are considered to be the best alternative when replacing the diesel internal combustion engine, especially in heavy vehicles where a battery system would be complicated and cumbersome (e.g., trucks and engineering machinery vehicle) [10].

In addition to cost, the application of PEMFCs in forklifts must contend with diesel internal combustion engines and charging batteries from the perspective of efficiency, stability, and durability [11]. This is because forklifts are constantly subjected to acceleration, deceleration, reversal, and lifting. These dynamic load cycles not only increase energy consumption but also irreversibly shorten the lifespan of PEMFCs due to PEM degradation and catalyst agglomeration [12,13]. Therefore, it is necessary to optimize the configuration of PEMFC systems to supplement the increased energy consumption and lifespan decay in specific application scenarios. 

Learning from hybrid vehicles (which combine combustion engines and batteries), a good engineering idea that combines PEMFCs and batteries was also proposed, where PEMFCs and batteries behave as complementary partners rather than competitors [14]. The batteries can discharge under high loads, and then can be charged by the PEMFCs, or by the recycling energy when braking, which achieves higher energy utilization (system efficiency) than the forklift installed with a pure PEMFC system. Beyond this, the instantaneous high-power demand can be replenished by the batteries, which could reduce the voltage fluctuation of PEMFCs, thus slowing down the degradation of PEMFCs caused by cyclic voltage and further increasing the maneuverability of a forklift installed with a PEMFC system [15,16,17]. 

As for research about application of PEMFC forklifts, most research focuses on the thermal, fluid modeling simulation calculations and model analyses [18,19,20,21,22], where the usual auxiliary batteries were lead-acid batteries [23,24,25]. The actual running data of forklifts that are powered by a PEMFC-based system (with lithium-ion batteries as auxiliary batteries) are deficient at present. In the start-up stage of PEMFC forklifts, the main focus should be comparisons with traditional power sources such as internal combustion engines, lead-acid batteries and lithium-ion batteries. As well, the economy and stability of PEMFC forklifts should be taken into consideration. The price of hydrogen, the cost of a PEMFC system, and the safety design are the main uncertainties that determine the cost competitiveness of PEMFCs [26]. Except for resolving technical difficulties, the corresponding government policies that support the development of the whole supply chain are also of great importance in promoting the development of this field. 

In this paper, a PEMFC power system was made based on the Zoomlion FB35Z 3.5 tonne electric counter-balance forklift. To ensure the PEMFC-powered forklift could adapt to different application environments, a homemade liquid-cooled PEMFC stack was integrated into the PEMFC power system that was combined with lithium-ion batteries to test the economic and work efficiencies. The handling capacity and mileage of the PEMFC forklift with a full tank of hydrogen was investigated; the operational cost and cost recovery cycle was also compared to that of a diesel internal engine forklift.

## 2. PEMFC/Lithium-Ion Battery Hybrid System Design

The homemade PEMFC stack was first evaluated on the Rigor fuel cell test system (RG24020) at 75 °C, with 80% relatively humidity (RH) under different backpressure. The backpressure of H_2_ and air (air stoichiometric ratio: 2.5) were first set at 140 kPa and 130 kPa, respectively. The polarization curve of the stack displayed a typical pattern and reached an output power of 15.8 kW. After decreasing the backpressures of H_2_ and air to 60 kPa and 50 kPa (air stoichiometric ratio: 1.8), respectively, there was a decrease in stack performance (Figure 1), where the current experienced a decrease from 270 A to 190 A at a stack voltage of 58.5 V, with a sharp drop in output power from 15.8 kW to 11.11 kW.

Figure 2a shows the PEMFC hybrid power system, which has an integrated a PEMFC system and lithium-ion batteries. The PEMFC system integrated the PEMFC stack and three subsystems. The air supply subsystem consisted of an air pump, a humidifier, an air filter, a restrictor, etc. The thermal management subsystem included a water pump, a radiator fan, a water tank, a deionizer, etc. The hydrogen supply subsystem included a hydrogen cylinder, a cylinder valve, a relief valve, a solenoid valve, a circulation pump, a steam separator, a vent valve, etc. The output voltage after the main direct current direct current converter (DCDC) was 85 V. The charge voltage of the lithium-ion batteries was 80 V and the lithium-ion batteries were connected to the DCDC bus. The auxiliary parts such as the fans, the water pump, the fuel cell control unit (FCU) were connected to the main DCDC converter through a 24 V DCDC converter. Figure 2b shows the refueling of the PEMFC forklift. The detailed parameters of the PEMFC power system are presented in Table 1.

The schematic of the PEMFC power system is presented in Figure 3. The purple line represents the signal, the red and black line represent electricity, the green line represents hydrogen, the blue line represents air, and the light blue line represents the coolant. The electricity output by the PEMFC power system was allocated either to the drive motor (maximum power: 12 kW) or the lifting motor (maximum power: 14 kW). The CAN bus has been employed to communicate to each of the auxiliary parts. Data for the state of charge (SOC) of the battery, the voltage of the PEMFC stack, the hydrogen pressure, the air flow, the temperature, the FCU monitors and coordinates of the auxiliary parts that controlled the output power, and the start and stop of the PEMFC power system were collected. The main functions of the FCU included the power-on and power-off of the PEMFC power system, the work mode, the control of auxiliary parts, the fault diagnosis and processing, the torque control, etc.

Since the response of the current output is in the microsecond range, air starvation would occur during drastic load fluctuation without a load change prediction, due to hysteresis of the air pump. In order to avoid this, when the FCU received signal to increase the output power, the FCU sent signals to the air pump and valves to responsd in advance, initializing an increase in the output current when the flow of the reactors reaches the demand. The electricity was converted to 80 V by the main DCDC, and was supplied to the motor through a high voltage box and a high voltage management unit. The lithium-ion batteries were charged when the output power of the PEMFC stack was greater than the power consumption of the forklift. In the contrast, the lithium-ion batteries were discharged when the output power of the PEMFC stack was insufficient to thus meet the demand for the instantaneously larger power consumption of the forklift. The PEMFC system stopped working when the SOC of the lithium-ion battery was greater than 90%, wherein the lithium-ion batteries became the main power to drive the motor. The PEMFC system started to work when the SOC of the lithium-ion batteries was less than 30% untill the SOC reached 90%. The main power dissipation of the PEMFC system was the air pump and radiator fan, while that of the water pump and valves was negligible. 

Figure 4 shows the I-E curves of the PEMFC stack in the power system. Hydrogen and air humidity cannot be controlled directly in a real PEMFC system. The humidification of hydrogen is achieved by mixing the wet hydrogen from the outlet of the stack and dry hydrogen by a hydrogen circulation pump. The humidity of the air was manipulated by a humidifier through the control of flow and temperature. The output current was 160 A when the voltage of the stack was 55.26 V, reaching an output power of 8.84 kW. Compared to the performance of the PEMFC stack with the backpressure of air at 50 kPa, the voltage of the stack decreased by 6.1 V. 

Since the air pressure ratio was only 1.2, the flow of air was greatly decreased by the fluid resistance of the air supply sub-systems (stack, humidifier, restrictor, tubes, etc.). The maximum flow rate was measured to be 49.96 kg/h, at which the maximum output currents of the PEMFC stack were 171 A and 238 A under air stoichiometric ratio of 2.5 and 1.8, respectively. However, there was a pressure drop caused by the humidifier, air filter, and stack, which caused a decrease in air back-pressure to a level lower than 10 kPa, and eventually contributed to a degradation of the PEMFC stack performance. The single cell voltage was close to 0.6 V when the PEMFC stack output current was 160 A. The hydrogen consumption rate increased when if the single cell discharged below 0.6 V. Therefore, a corresponding air stoichiometric ratio of 2.68 was applied to maintain a higher discharging voltage (V_single cell_ > 0.6 V) for better energy conversion efficiency. The I-E curve of the PEMFC stack was converted to the polarization curves of a single fuel cell. As shown in Figure 4, when the single cell voltage was 0.6 V, the output current density of the stack reached 1.1 A/cm^2^.

The single cell voltage fluctuation ratio is the relative standard deviation of single cell voltage, and its calculation formula is shown in Formula (1).
(1)$$S_{r}=\sqrt{\overset{N}{\underset{j=1}{\sum }}{\left(V_{j}-\bar{V}\right)}^{2}/\left(N-1\right)/\bar{V}}$$
where N is the number of single cells, V_j_ is the voltage of a single cell, and $\bar{V}$ is the average voltage of single cells. The root mean square value of voltage variation in a single cell is the average voltage deviation of each single cell from the average voltage, which reflects the dispersion degree of single cell voltage. Generally, larger voltage fluctuation of single-cell voltage indicate worse consistency in a single cell, and ultimately poor stability of the PEMFC stack. Its mathematic expression is shown as Formula (2): (2)$$\sigma_{x}=\sqrt{\frac{1}{N}\overset{N}{\underset{j=1}{\sum }}{\left(V_{j}-\bar{V}\right)}^{2}}$$

To evaluate the consistency of single cells, the voltage of a single cell at 165 A was recorded. As shown in Figure 5a, the voltage volatility was determined to be 1.56%, while the root mean square value of the variation in single cell voltage was 1.2%. Interestingly, the average voltage of the PEMFC stack was 0.617 V, whereas that of the outmost membrane electrode assemblies (MEAs) (the 1st and 90th MEA) were 0.559 V and 0.556 V, respectively. The low voltage at the boundary regions could be related to their high heat dissipation rate, which resulted in the lower temperatures at the two ends compared to in the middle of the stack. The decreased catalyst activity and membrane proton conductivity at low temperatures suppressed reactions, and thus yielded lower single-cell voltages. 

In the practical operation of the PEMFC stack, differences in batch quality and operation status of MEAs may result in different lifetimes of MEAs. Replacement of those degraded MEAs is necessary to ensure the stable operation of the PEMFC stack. In order to investigate the influence of replacing old MEAs with new MEAs on the performance of PEMFC stack, a stack was assembled using different batches of used MEAs. As shown in Figure 5b, the voltage of the 2nd, 13th, 37th, 55th, 57th, 63th, 64th and 90th MEA in the PEMFC stack were lower than the average voltage. The eight pieces were replaced with new MEAs, while the 43rd and 89th MEAS were replaced for reference. As shown in Figure 5b, the voltages of the replaced MEAs were still lower than the average voltage after reassembling, and their performance was even worse than that of MEAs before replacement. The single cell voltage volatility increased from 3% to 5.73%, while the root mean square value climbed from 2.34% to 4.45%. The increased degree of dispersion may be ascribed to the fact that these 10 pieces of new MEAs were not activated before use. The activation degree of the replaced MEAs was not comparable to the rest of the pieces, which caused a lower voltage than the average values, indicating the necessity of MEA activation prior to the replacement of old MEAs.

Water management of the PEMFC stack was necessary to prevent flooding of the catalyst layer, while the accumulation of inert gases also required timely venting during operation. Since an improper venting cycle would lead to fierce fluctuation of the voltage, the interval of hydrogen venting (1800 A·s) was set considering both time and current. It can be seen from the chronoamperometry (Figure 6) that there were two platform areas during testing. The startup of the PEMFC stack (feeding of gas) first triggered a jump in voltage to 85 V. Afterwards, a stepwise discharge was applied to humidify the membrane as well as to activate the catalyst. After activation of the MEAs, the discharge current of the PEMFC stack was fixed at 130 A, whose corresponding voltage was stabilized at 59.6 V with a fluctuation in voltage of less than 0.3 V. Similar stability was also witnessed after increasing the current to 160 A, but it reached a slightly lower voltage platform at 55.2 V. Stable discharging is a critical factor that affects the lifetime of the PEMFC stack, where unstable voltage in constant-current discharge accelerate the breakdown of MEAs and other parts, while fluctuated current under fixed voltage is more prone to cause poor stability in output power. As they are closely involved in the operation of the stack, coolant temperature and air flow are of great importance as well. 

As shown in Figure 6, 42.9 kg/h and 49.2 kg/h air flow were applied to obtain the desired discharging currents of 130 A and 160 A, respectively. The variation in air flow and slight increase in coolant temperature matched well with the chronoamperometry diagram, demonstrating a good connection between subsystems that provided the PEMFC stack with excellent controllability. These stable operation parameters demonstrated the good qualities of the MEAs and the PEMFC stack. The minor impact of flooding confirmed the feasibility of the venting cycle.

The PEMFC stack was subsequently installed in a Zoomlion counterbalance forklift (FB35Z, lifting capacity: 3.5 tonne) for field testing. The on-site test was conducted according to the following standards that involved four stages until exhausting all of the hydrogen:(1)Lifting 1 tonne load to 2.5 m and then lowering it to 0.3 m;(2)Driving the loaded forklift at a distance of 160 m;(3)Repeating stage (1);(4)Driving the loaded forklift back to the starting point.

Figure 7 compares the work efficiency of the PEMFC-powered forklift (hydrogen pressure: 33 MPa) and the lithium-ion battery-powered forklift (Linde counterbalance forklift, lifting capacity 3.5 tonne, battery capacity: 33 kWh). The cut-off values for hydrogen pressure and voltage were set to 3 MPa and 20% SOC. As we can see in Figure 7, the mileage of the PEMFC forklift was about 22.5 km and the run time was about 150 min, equal to 140.6 times transport and 1.067 min/cycle. By contrast, the lithium-ion battery-powered forklift had a milage of 25.7 km and a run time of 229 min, whose 160 times transport and 1.431 min/cycle indicated that the PEMFC-powered forklift has better working efficiency than the lithium-ion battery-powered forklift. Assuming the working time of two forklifts is 8 h, the lithium-ion battery-powered forklift took 30 min to change the batteries every time, which would mean a total working time of 450 min and a 310 times transport. In contrast, the PEMFC powered forklift needed refueling 3 times, and its fast refueling (only 5 min) would permit a total working time of 465 min and a 435.8 times transport. The working efficiency of the PEMFC-powered forklift was 40.6% higher than that of the lithium-ion battery-powered forklift. 

Figure 8 schematically presents the self-constructed hydrogen station which combines hydrogen production, storage and refueling inside the factory. The solar-driven hydrogen production was built by connecting solar panels and a PEM electrolyzer. The as-produced high purity hydrogen (99.99%) was then compressed and pressurized into the hydrogen tank (maximum capacity at 20 MPa: 700 Nm^3^) from 1.3 MPa to 20 MPa by a gas-driven hydrogen compressor. The inlet of the hydrogen storage tank was equipped with a stop valve, which was used to block the hydrogen of the compressor from entering the hydrogen storage tank in the case of a leakage. At the same time, a pressure relief valve was installed in the hydrogen storage tank, which was used to discharge overcharged hydrogen when the hydrogen pressure exceeded the set pressure. The hydrogen in the storage tank was further pressurized to 45 MPa by an electric compressor in the mobile hydrogen refueling station. Additionally, during refueling, high pressure hydrogen was filled into the 35 MPa carbon fiber-wound aluminum alloy tank of the PEMFC-powered forklift.

The water electrolyzer worked 16 h a day, during which an 8 h power supply came from the photovoltaic plant (the light coefficient is 3.4 in Foshan, China). The power from another 8 h power supply came from an electric power supply during mid-night, whose price was 4 cent per kilowatt. 

Assuming the prepared hydrogen is supplied to 20 forklifts, which work for 8 hours a day and 250 days a year (excluding weekends and legal holidays), each forklift needs 3.06 kg of hydrogen per day. The daily hydrogen consumption of 20 forklifts will be 61.2 kg, and the hydrogen production rate of the electrolytic water hydrogen production system should exceed 42.84 Nm^3^/h. The total cost of all of the equipment which includes the water electrolyzer, hydrogen storage tank, mobile refueling station and PEMFC forklift is about USD 1.437 million. The cost of the PEMFC stack is about 15.6% of the total cost of the PEMFC-powered forklift. The cost of the PEMFC powered forklift accounts for about 60% of the total cost of all equipment.

The use-cost of this scheme can be subdivided into the expenditures on water and electricity during the valley period, the annual use-cost was estimated to be~19,466 dollars per year. All of the facilities were esimated to depreciate according to a 10-year service life. The total cost of hydrogen production and hydrogenation would be about USD 337,420, including the water electrolyzer, the hydrogen storage tank and the mobile hydrogen refueling station. Considering that maintenance costs are 2% of the total cost of hydrogen production and the hydrogenation equipment, the total equipment cost is USD 344,168.4. That means the total cost of hydrogen production for 10 years is about USD 538,828.4. The total hydrogen consumption of the forklifts for 10 years is 153,000 kg, so the cost of hydrogen is about USD 3.52 per kilogram. Table 2 shows the fuel cost of the forklift with different power systems.

Compared with the diesel engine forklift, the annual fuel cost saved by the PEMFC forklift with this scheme is USD 289017. Considering the cost saved by fuel, the total cost of the PEMFC forklift, the full set of hydrogen production and the refueling system can be recovered in 4.9 years. The total cost of the forklifts (the equipment cost, the maintenance cost, and the operating cost) driven by two different power sources for 10 years was compared. The total cost of the PEMFC forklift over 10 years is USD 1631600, while that of the diesel engine forklift over 10 years is USD 3610045.6. Twenty PEMFC forklifts can save USD 1978385.6 over 10 years and reduce emissions of carbon dioxide by about 6838 tonne compared to 20 diesel internal combustion engine-powered forklifts. 

## 3. Conclusions

The PEMFC hybrid power system combining a PEMFC system and lithium-ion batteryies was tested on a forklift. The venting cycle considering both current and time minimized the fluctuation of voltage. The work efficiency and mileage of the PEMFC-powered forklift and diesel engine forklift were compared, with the result showing that the PEMFC powered forklift has higher work efficiency and longer mileage. The comparison of economic efficiency of the two different power systems showed that the PEMFC-powered forklift has great advantages in terms of the use-cost. A remarkable reduction in exhaust gas emissions is not only significant for environmental protection, but also brings great benefits for customers in future carbon trading.

## Figures and Tables

**Figure 1 molecules-27-04918-f001:**
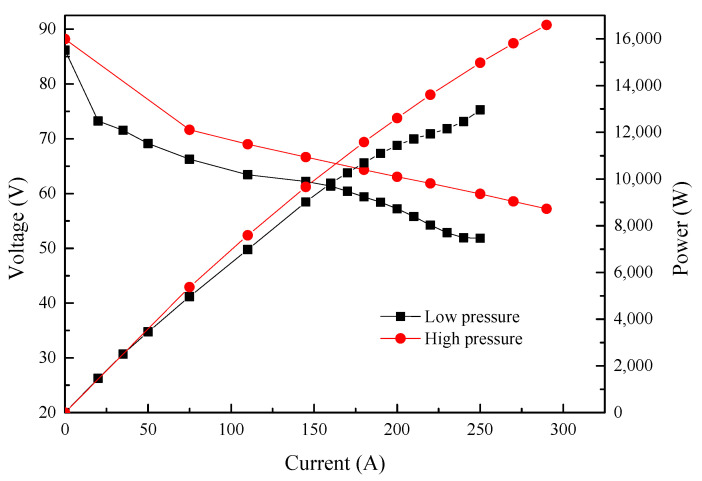
Performance of the PEMFC stack.

**Figure 2 molecules-27-04918-f002:**
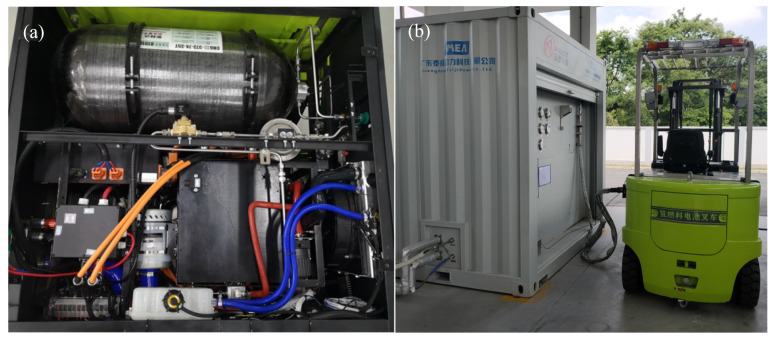
Fuel cell system and Refueling for the PEMFC powered forklift.

**Figure 3 molecules-27-04918-f003:**
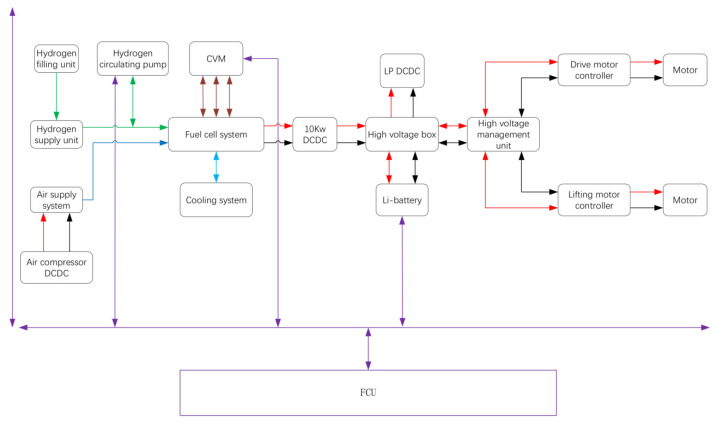
Overview of PEMFC hybrid drive system.

**Figure 4 molecules-27-04918-f004:**
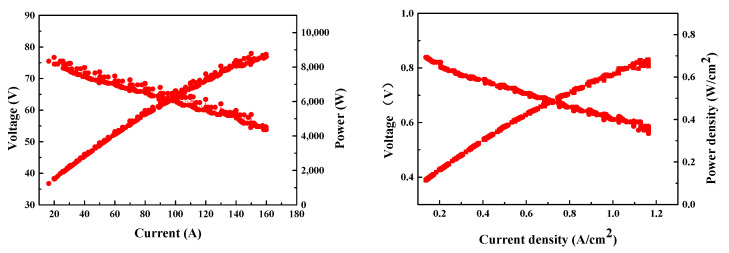
Performance of PEMFC in the fuel cell system (test condition: H_2_/air stoichiometric ratio: 1.2/2.68, H_2_/air back-pressure: 35/8 kPa).

**Figure 5 molecules-27-04918-f005:**
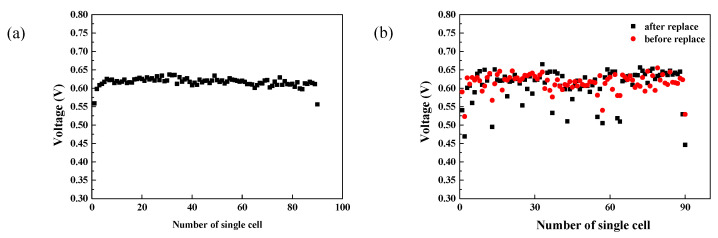
The cell voltage of a normal PEMFC stack with high consistency (**a**) and the cell voltage of a PEMFC stack before and after replacing specific cells (**b**).

**Figure 6 molecules-27-04918-f006:**
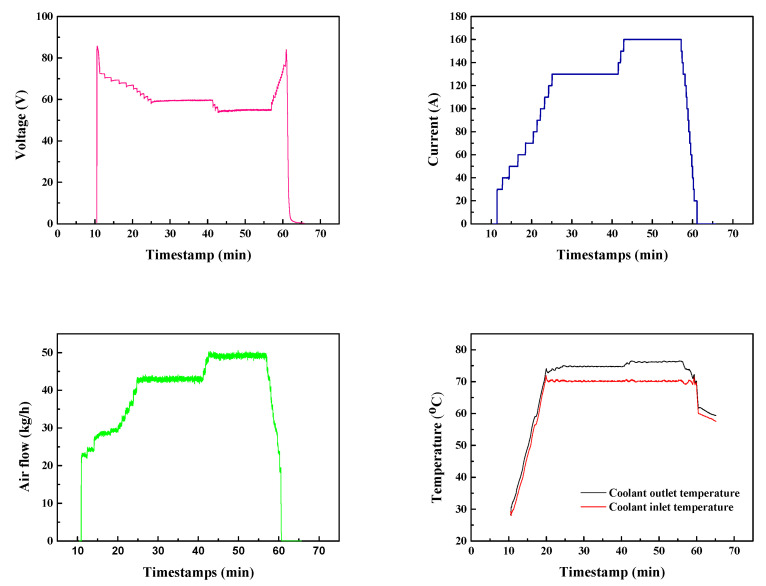
Fuel cell stack voltage variation and operation parameters of the PEMFC system.

**Figure 7 molecules-27-04918-f007:**
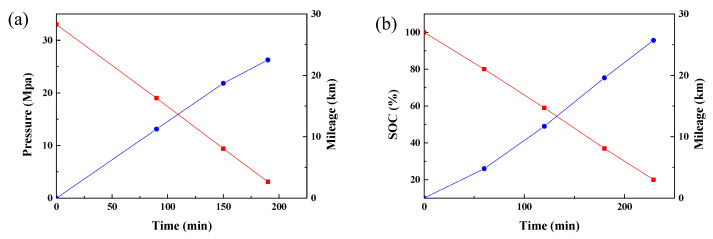
The mileage of the PEMFC forklift (**a**) and lithium ion battery forklift (**b**).

**Figure 8 molecules-27-04918-f008:**
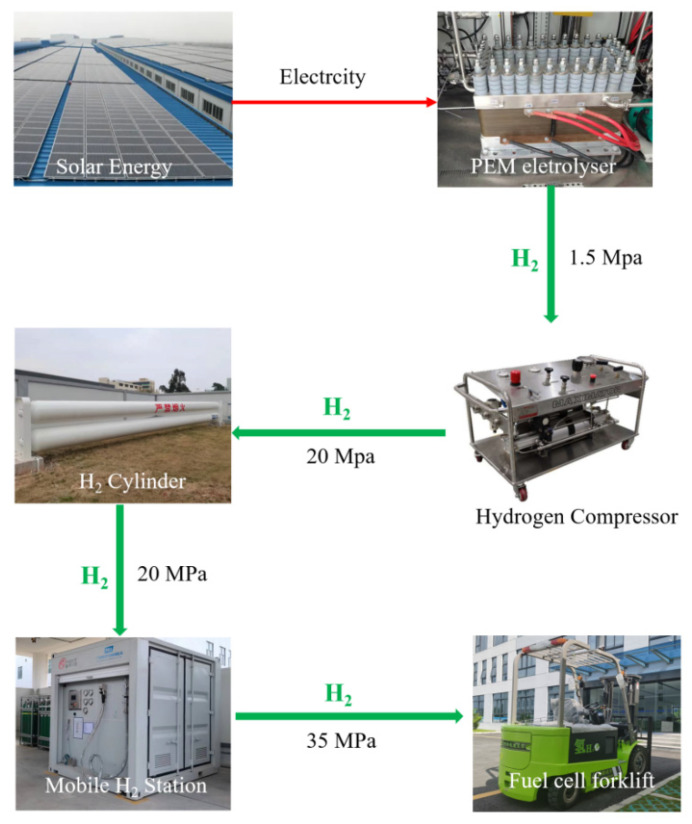
Integrated hydrogen production and hydrogenation station.

**Table 1 molecules-27-04918-t001:** Parameters of the PEMFC power system.

Parameters	Value
Number of Cells in Stack	90
Maximum output Current of stack	165 A
Rated Voltage of Stack	54~72 V
Output Voltage of PEMFC system	80 V
Maximum Output Power of system	32 kW
Size of whole system	1124 × 1018 × 564 mm
Weight of PEMFC system	400 kg

**Table 2 molecules-27-04918-t002:** The fuel cost of 20 forklifts with different power system.

Forklift	Power System	Fuel Cost	Total Fuel Cost per Year
3T	Diesel engine	USD 1.3187 /L	USD 342,900
3.5T	Fuel cell	USD 3.52 /kg	USD 53,882.8

## Data Availability

Not available.

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
