# Peer review of "Work Efficiency and Economic Efficiency of Actual Driving Test of Proton Exchange Membrane Fuel Cell Forklift"

_molecules, 2022, doi:10.3390/molecules27154918_

Round 1
Reviewer 1 Report
In this manuscript the authors investigated the work efficiency and economic efficiency of a PEM fuel cell under an actual working condition. It is a nice work from the practical point of view though the authors should pay attention in the figure captions. In general, additional descriptions should be added in the caption. For example, in Figure 3, different colors were used in the arrows, authors should explain it and the give a brief introduction of how the PEMFC hybrid drive system works. Additionally, authors should be cautious regarding the significant digits. E.g., line 276 in Table2.
Author Response
Dear Reviewer #1
We appreciate your valuable comments and enlightenment rendered to our manuscript.
Your specific questions raised are addressed as follows.
Q1: In this manuscript the authors investigated the work efficiency and economic efficiency of a PEM fuel cell under an actual working condition. It is a nice work from the practical point of view though the authors should pay attention in the figure captions. In general, additional descriptions should be added in the caption. For example, in Figure 3, different colors were used in the arrows, authors should explain it and the give a brief introduction of how the PEMFC hybrid drive system works.
Response: Thanks for your valuable comments. We have updated the figure 3 to make it more clear to readers and added explanation “The purple line represents signal, the red and black line represent electricity, the green line represents hydrogen, the blue line represents air, the light blue line represents coolant.”
Q2: Additionally, authors should be cautious regarding the significant digits. E.g., line 276 in Table2.
Response: Thanks for pointing out the errors. We made the mistake in converting RMB to dollars. We have revised the digits. Considering the maintenance cost of 2% of the total cost of hydrogen production and hydrogenation equipment, the equipment cost is 344168.4 dollars. That means the total cost of hydrogen production for 10 years is about 538828.4 dollars. The total hydrogen consumption of the forklifts for 10 years is 153000 kg, so the cost of hydrogen is about 3.52 dollars per kilogram. Table 2 has shown the fuel cost of forklift with different power system.
Table 2. The fuel cost of 20 forklifts with different power system.
Forklift Power system Fuel cost Total fuel cost per year
3T Diesel engine 1.3187 dollars/L 342900 dollars
3.5T Fuel cell 3.52 dollars/kg 53882.8 dollars
Compared with the diesel engine forklift, the annual fuel cost saved by the PEMFC forklift with this scheme is 289017 dollars. Considering the cost saved by fuel, the cost of PEMFC forklift and the full set of hydrogen production and refueling system can be recovered in 4.9 years. The total cost of forklifts (equipment cost, maintenance cost and operating cost) driven by two different power source for 10 years has been compared. The total cost of PEMFC forklift in 10 years is 1631600 dollars, while that of diesel engine forklift in 10 years is 3610045.6 dollars. Twenty PEMFC forklifts can save 1978385.6 dollars over 10 years and reduce emission of carbon dioxide about 6838 tons compared to 20 diesel internal combustion engine forklifts.
Once again, on behalf of the co-authors, I would like to express my gratitude to you for the precious time spent and the enlightenment.
Yours faithfully,
Zi’ang Xiong
Reviewer 2 Report
In this manuscript, the forklift comprising PEMFC and lithium-ion battery has been manufactured, tested, and compared. It is found that the back-pressure of air is closely related to the power density, and the PEMFC stability could be improved by a reasonable control strategy and MEAs with high consistency. PEMFC-powered forklift displayed higher work efficiency than lithium-ion batteries powered forklift. Its lower use-cost compared to internal engines powered forklifts are beneficial to commercialization.
I may give a minor revision due to further improvements that are needed by addressing my comments made below.
(1) For the Keywords, I suggest adding ‘lithium-ion battery’, ‘power density’, ‘control strategy’, and ‘membrane electrode assemblies’ to attract a broader readership.
(2) There are certain grammar problems. It is suggested that the full text be improved a bit.
(3) Line 31, ‘One solution to this customer demand is developing electric forklift that combines power source and electric motor.’ It is suggested to combine the global era background - that is, the goal of decarbonization, so electricity can be accessed through renewable energy, so as to avoid the traditional internal combustion engine still using a lot of fossil fuels and emitting greenhouse gases. However, renewable energy sources are unstable and intermittent during generation, and thus these valuable electric energies are difficult to apply continuously and stably. This may open spatial and temporal gaps between the availability of the energy and its consumption by the end-users.To tackle this issue, the employment of energy storage systems may greatly improve the utilization rate and stability of renewable energy [ChemSusChem 15.1 (2022): e202101798; Electrochimica Acta 309 (2019): 311-325].
(4) Line 70, ‘To our best knowledge, the realistic test data about work efficiency and cost of PEMFC powered forklift have not been reported yet.’ I do not agree with this sentence, and some brief introduction about these references should be added [10.1021/acsomega.1c07344; 10.1149/MA2020-02342185mtgabs].
(5) For Table 1, about the PEMFC system, which membrane is used, and how about the type of catalysts adopted? Normally, the membrane used should be Perfluorosulfonic Acid Membranes, which have very good proton conductivity but also a very high cost [Electrochimica Acta 378 (2021): 138133]. This information is directly related to the capital cost of such a system, and further details should be provided.
(6) Line 166, ‘Figure 5. The cell voltage of the PEMFC stack and the cell voltage of the PEMFC stack before and after replaced specific cells’. I consider ‘the cell voltage of PEMFC’ should only appear once, not twice.
(7) Why in the cost comparison, the lithium-battery cost is not considered together with PEMFC? That will also increase the total cost of the PEMFC-based system compared to the internal combustion system.
Author Response
Dear Reviewer #2
We appreciate your valuable comments and enlightenment rendered to our manuscript.
Your specific questions raised are addressed as follows.
Q1: For the Keywords, I suggest adding ‘lithium-ion battery’, ‘power density’, ‘control strategy’, and ‘membrane electrode assemblies’ to attract a broader readership.
Response: Thanks for your valuable comments. Due to the limit of numbers of keywords, we have added “control strategy”, “power density” and “membrane electrode assemblies”.
Q2: There are certain grammar problems. It is suggested that the full text be improved a bit.
Response: Thanks for your kind advice. We have made attempts to improve the quality of the English in the revised manuscript.
Q3: Line 31, ‘One solution to this customer demand is developing electric forklift that combines power source and electric motor.’ It is suggested to combine the global era background - that is, the goal of decarbonization, so electricity can be accessed through renewable energy, so as to avoid the traditional internal combustion engine still using a lot of fossil fuels and emitting greenhouse gases. However, renewable energy sources are unstable and intermittent during generation, and thus these valuable electric energies are difficult to apply continuously and stably. This may open spatial and temporal gaps between the availability of the energy and its consumption by the end-users.To tackle this issue, the employment of energy storage systems may greatly improve the utilization rate and stability of renewable energy [ChemSusChem 15.1 (2022): e202101798; Electrochimica Acta 309 (2019): 311-325].
Response: Thanks for your valuable comments. We have added the introduction about goal of decarbonization and demand for stable and renewable energy.
Q4: Line 70, ‘To our best knowledge, the realistic test data about work efficiency and cost of PEMFC powered forklift have not been reported yet.’ I do not agree with this sentence, and some brief introduction about these references should be added [10.1021/acsomega.1c07344; 10.1149/MA2020-02342185mtgabs].
Response: Thanks for your valuable comments. We have deleted that sentence and cited this two references.
Q5: For Table 1, about the PEMFC system, which membrane is used, and how about the type of catalysts adopted? Normally, the membrane used should be Perfluorosulfonic Acid Membranes, which have very good proton conductivity but also a very high cost [Electrochimica Acta 378 (2021): 138133]. This information is directly related to the capital cost of such a system, and further details should be provided.
Response: Thanks for your valuable comments. The catalyst is Pt/C of 60wt.% which supplied by Chinese manufacturer. The proton exchange membrane is provided by Chinese supplier which the name is Dongyue group. The cost of PEMFC stack is about 6900 dollars when the yield of stack is 100 pcs, which is only about 15.6% of the total cost of PEMFC forklift. The cost of MEA accounts for the 36% of the total cost of PEMFC stack. The cost of PEMFC stack could be reduced to 5000 dollars if the yield could reach to 5000 pcs per year. The cost of MEA stack is a small fraction of the total cost of whole system.
Q6:Line 166, ‘Figure 5. The cell voltage of the PEMFC stack and the cell voltage of the PEMFC stack before and after replaced specific cells’. I consider ‘the cell voltage of PEMFC’ should only appear once, not twice.
Response: Thanks for your valuable advice. We have revised it. The sentence has been changed to “ The cell voltage of normal PEMFC stack with high consistancy: a) and the cell voltage of PEMFC stack before and after replaced specific cells: b)”.
Q7:Why in the cost comparison, the lithium-battery cost is not considered together with PEMFC? That will also increase the total cost of the PEMFC-based system compared to the internal combustion system.
Response: Thanks for your kindly advice. The total cost of the PEMFC forklift included the cost lithium-battery, PEMFC system and bodywork of forklift.
Once again, on behalf of the co-authors, I would like to express my gratitude to you for the precious time spent and the enlightenment.
Yours faithfully,
Zi’ang Xiong